# Two-subunit DNA escort mechanism and inactive subunit bypass in an ultra-fast ring ATPase

**Ninning Liu[1,2†‡], Gheorghe Chistol[1,3†§], Carlos Bustamante[1,2,3,4,5,6,7]\***

[1]Jason L. Choy Laboratory of Single Molecule Biophysics, University of California, Berkeley, United States; [2]Department of Molecular and Cell Biology, University of California, Berkeley, United States; [3]Department of Physics, University of California, Berkeley, United States; [4]California Institute for Quantitative Biosciences, Berkeley, United States; [5]Department of Chemistry, Howard Hughes Medical Institute, University of California, Berkeley, United States; [6]Physical Biosciences Division, Lawrence Berkeley National Laboratory, Berkeley, United States; [7]Kavli Energy NanoSciences Institute at the University of California, Berkeley and the Lawrence Berkeley National Laboratory, Berkeley, United States

**\*For correspondence:** carlosb@berkeley.edu

[†]These authors contributed equally to this work

**Present address:** [‡]Wyss Institute for Biologically Inspired Engineering, Harvard University, Boston, United States; [§]Department of Biological Chemistry and Molecular Pharmacology, Harvard Medical School, Boston, United States

**Abstract** SpoIIIE is a homo-hexameric dsDNA translocase responsible for completing chromosome segregation in *Bacillus* subtilis. Here, we use a single-molecule approach to monitor SpoIIIE translocation when challenged with neutral-backbone DNA and non-hydrolyzable ATP analogs. We show that SpoIIIE makes multiple essential contacts with phosphates on the 5'→3' strand in the direction of translocation. Using DNA constructs with two neutral-backbone segments separated by a single charged base pair, we deduce that SpoIIIE's step size is 2 bp. Finally, experiments with non-hydrolyzable ATP analogs suggest that SpoIIIE can operate with non-consecutive inactive subunits. We propose a two-subunit escort translocation mechanism that is strict enough to enable SpoIIIE to track one DNA strand, yet sufficiently compliant to permit the motor to bypass inactive subunits without arrest. We speculate that such a flexible mechanism arose for motors that, like SpoIIIE, constitute functional bottlenecks where the inactivation of even a single motor can be lethal for the cell.

## Introduction

The ASCE [Additional Strand Conserved E (glutamate)] division of oligomeric, ring-shaped NTPases encompasses a diverse range of enzymes that function as molecular motors (*Lyubimov et al., 2011*). Within the ASCE division, the FtsK/SpoIIIE family of motors is involved in the fundamental process of DNA segregation prior to cell division. During the *Bacillus* subtilis sporulation lifecycle, an asymmetric division septum closes around one of the sister chromatids before complete chromosome segregation, trapping about two-thirds of the chromosome in the mother cell compartment (*Burton et al., 2007*; *Wu and Errington, 1994*). To complete chromosome segregation, SpoIIIE must translocate the DNA from the mother cell into the forespore. SpoIIIE and its closely related *Escherichia* coli homologue FtsK, contain an N-terminal transmembrane domain that anchors the protein to the division septum, a long unstructured polypeptide linker and a C-terminal-soluble motor domain consisting of subdomains α, β, and γ (*Barre, 2007*). Subdomains α and β adopt a RecA-like fold containing ATP binding and hydrolysis motifs (*Massey et al., 2006*), while subdomain γ imparts translocation directionality to the motor (sequence dependence) (*Besprozvannaya et al., 2013*; *Lee et al., 2012*; *Löwe et al., 2008*; *Ptacin et al., 2006*; *2008*). Crystallography and electron

**eLife digest** *Bacillus subtilis* is a bacterium that lives in the soil. When food is in short supply, *B. subtilis* stops reproducing and individual bacterial cells transform into spores that lay dormant until conditions improve. While, *B subtilis* is generally harmless, it forms spores in a similar way to other bacteria that cause diseases such as anthrax.

During spore formation, a membrane forms to divide the cell into a large mother cell and a smaller "forespore" cell. Then, a copy of the mother cell's DNA – which is made of building blocks called bases – moves into the forespore. A group of proteins called SpoIIIE is instrumental in this process as it uses energy from a molecule called ATP to pump the DNA across the membrane at the rapid speed of 5,000 base pairs of DNA per second. SpoIIIE contains six individual protein subunits that form a ring-shaped motor structure that spans the membrane. It belongs to a large family of proteins that are found in all living organisms and drive many vital processes.

How does SpoIIIE interact with DNA and how do the individual subunits coordinate their behaviour? Liu, Chistol et al. address these questions by using instruments called optical tweezers, which use a laser beam to hold and manipulate tiny objects. The experiments show that to move a fragment of DNA across a membrane, SpoIIIE only makes contact with one of the two strands that make up the DNA molecule. The experiments suggest that the DNA is handed over from one SpoIIIE subunit to another in a sequential order. This would allow the DNA to remain bound to SpoIIIE at all times as it passes through the membrane.

Next, Liu, Chistol et al. measured how SpoIIIE steps along the DNA and found that each subunit takes a small two base pair step when energy is released from a single molecule of ATP. There is an element of flexibility in the system, because SpoIIIE can still move DNA normally even if some subunits cannot use energy from ATP. This provides a fail-safe mechanism that still allows the cells to form spores in the event that one subunit is disabled. Future work will concentrate in understanding how the subunits communicate around the ring to coordinate their sequential use of ATP and their DNA pumping activity.

microscopy studies indicate that both FtsK and SpoIIIE form homo-hexameric rings, and that double-stranded DNA (dsDNA) is threaded through their central pore (*Cattoni, et al., 2014*; *Cattoni et al., 2013*; *Massey et al., 2006*).

A distinguishing characteristic of SpoIIIE/FtsK is their enormous translocation velocity (~5 kbp/s) and their ability to work against high forces (*Ptacin et al., 2008*; *Saleh et al., 2004*). Previous single-molecule studies of these motors focused primarily on investigating the mechanism of sequence recognition and translocation direction reversal (*Lee et al., 2012*; *Pease et al., 2005*; *Ptacin et al., 2006*; *2008*; *Saleh et al., 2004*), studying how they strip off DNA-bound proteins (*Lee et al., 2014*; *Marquis et al., 2008*), and determining the amount of supercoils introduced in the DNA during translocation (*Saleh et al., 2005*). However, many fundamental aspects of these motors' operation remain poorly understood: How does the motor interact with its DNA track during translocation? What is the motor step size? How does the motor coordinate the activity of its individual subunits? How is the subunit coordination mechanism optimized for the motor's specific biological task?

To answer these questions, we used single-molecule manipulation and measurement techniques. Using modified DNA with a neutral backbone, we show that SpoIIIE makes critical electrostatic contacts with the phosphate backbone on the 5′→3′ strand in the direction of translocation. This observation indicates that the individual subunits operate in a well-defined sequential order around the ring. To determine the SpoIIIE step size, we challenged the motor to translocate a DNA molecule containing two neutral segments of variable lengths separated by a charged base pair. This hybrid construct revealed the periodicity of motor–DNA interactions, suggesting that each SpoIIIE subunit takes a 2-bp step per ATP hydrolyzed. Experiments where non-hydrolyzable nucleotides were used to probe the intersubunit coordination within the motor suggest that SpoIIIE can tolerate non-consecutive inactive subunits, implying a degree of flexibility in the sequential operation of the motor. Finally, we propose a two-subunit DNA escort model that can rationalize all these data and that correctly predicts the degree of supercoiling introduced by SpoIIIE during translocation.

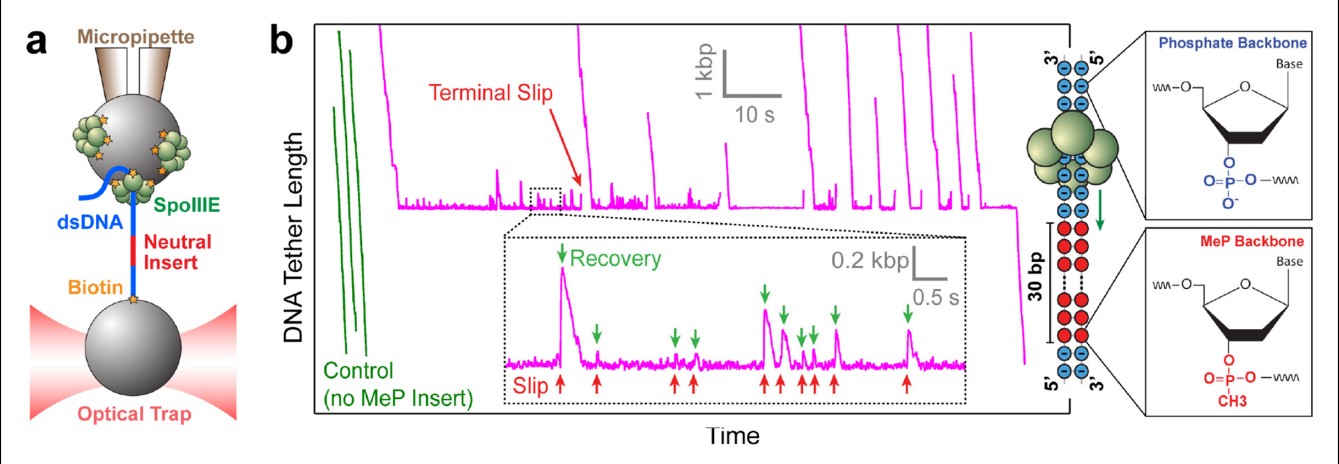

**Figure 1.** Probing SpoIIIE-DNA interactions with neutral DNA inserts. (a) Experimental geometry of the optical tweezers assay. Neutral MeP DNA (red) was placed roughly 4 kb away from the end of the dsDNA tether (blue) that is attached to the optically trapped bead. (b) Sample traces of individual SpoIIIE motors translocating on DNA with a 30-bp dsMeP insert (magenta). Control experiments with DNA containing no MeP inserts are shown in green. Inset: detailed view of motor's attempts to cross the neutral insert. All experiments were conducted at 3 mM ATP and a constant force of 5 pN. Traces were offset vertically to line-up the pause-like regions. Phosphate and MeP groups are represented as blue and red circles respectively.

The following figure supplements are available for Figure 1:

**Figure supplement 1.** A single SpoIIIE ring translocates DNA in single-molecule experiments..

## Results

To monitor DNA translocation by SpoIIIE, we used a single-molecule assay developed previously (*Smith et al., 2003*). Briefly, a DNA-coated bead was held in an optical trap while a SpoIIIE-coated bead was held on a micropipette (*Figure 1a*); SpoIIIE engaged the DNA when the beads were in close proximity, and translocated the DNA only in the presence of ATP. The DNA was held under constant tension by continuously adjusting the trap position as translocation proceeded. At saturating [ATP] (3 mM) and a low force (5 pN) SpoIIIE translocated DNA at ~4 kbp/s (*Figure 6—figure supplement 1*), in agreement with previously reported rates *in vivo* (*Burton et al., 2007*) and *in vitro* (*Ptacin et al., 2008*). Control experiments indicate that a single SpoIIIE motor was attached to the DNA tether: (i) Protein concentration was titrated until activity was observed for only one of every ~5 pairs of beads tested, thus reducing the likelihood that a single bead contained multiple active SpoIIIE motors. (ii) At high forces, SpoIIIE loses its grip on DNA and slips. These slipping events occurred in one fast rip, not in several steps, indicating that a single motor–DNA interaction is disrupted in each case (*Figure 1—figure supplement 1*).

### SpoIIIE makes critical contacts with the DNA phosphate backbone during translocation

Modified inserts such as ssDNA, dsDNA with interstrand cross-links, dsDNA with abasic sites, and so on, have been previously used in single-molecule experiments to probe how dsDNA translocases interact with their nucleic-acid track (*Aathavan et al., 2009*; *Stanley et al., 2006*). To investigate how SpoIIIE interacts with its substrate, we designed DNA constructs containing a modified insert with a methyl-phosphonate (MeP) backbone (*Figure 1b*), which preserves the base pairing and overall structure of B-form DNA (*Strauss and Maher, 1994*). We first monitored SpoIIIE translocation along a substrate containing a 30-bp double-stranded MeP (dsMeP) insert, several times longer than the expected step size of the motor. Upon reaching the insert, SpoIIIE undergoes multiple slips followed by translocation recoveries (*Figure 1b* inset), revealing that it makes several attempts to cross the neutral segment. Despite these attempts, SpoIIIE failed to traverse the 30-bp dsMeP segment (*Figure 1b*), indicating that motor interactions with the negatively charged phosphates are critical for translocation.

**Table 1.** MeP traversal statistics.

| DNA construct | Successful crossings | Failed crossings | Total traces |
|---|---|---|---|
| ssMeP modification | | | |
| 30 base 3'→5' MeP | 23 | 1 | 24 |
| 30 base 5'→3' MeP | 6 | 22 | 28 |
| dsMeP Modification | | | |
| 2 bp dsMeP | 21 | 1 | 22 |
| 3 bp dsMeP | 16 | 0 | 16 |
| 4 bp dsMeP | 21 | 3 | 24 |
| 5 bp dsMeP | 13 | 13 | 26 |
| 7 bp dsMeP | 3 | 9 | 12 |
| 10 bp dsMeP | 1 | 19 | 20 |
| 30 bp dsMeP | 1 | 31 | 32 |
| Length of MeP Probe Segment* | | | |
| 4 bp dsMeP probe | 7 | 16 | 23 |
| 3 bp dsMeP probe | 9 | 16 | 25 |
| 2 bp dsMeP probe | 6 | 18 | 24 |
| 1 bp dsMeP probe | 33 | 4 | 37 |

All MeP data was gathered under 5 pN of opposing force and [ATP] = 3 mM

*All MeP probe segments carry a 4 bp dsMeP modification upstream of the probe

Depending on the step size of the motor, and the manner in which it interacts with the DNA, the motor should be able to easily traverse short dsMeP inserts up to some critical length. To determine this length, we designed dsMeP inserts of varying size and recorded the motor's ability to cross each insert (*Figure 2a*). We found that SpoIIIE can traverse dsMeP segments of 2 bp, 3 bp, and 4 bp with near-100% probability (*Figure 2a*), but the traversal probability sharply drops to ~50% for dsMeP inserts of 5 bp (*Figure 2b*). Importantly, this sharp reduction in insert crossing probability is accompanied by a dramatic increase in the average traversal time (*Figure 2c*). Note that further increasing the dsMeP insert length from 5 bp to 7 bp does not significantly increase the mean traversal time (*Figure 2c*). Taken together, these data show that 5 bp is the minimum length of dsMeP required to disrupt processive translocation by SpoIIIE, and indicate that SpoIIIE makes periodic electrostatic contacts with the DNA every 5 bp or less.

Note that in rare instances (1 out of 32 molecules, *Table 1*), SpoIIIE managed to traverse the 30-bp dsMeP insert, albeit after several seconds of repeated crossing attempts (*Figure 1b*). A slightly higher traversal probability (~8%) was recorded for the 10-bp dsMeP insert (*Figure 2b*). The dynamics of slipping and re-translocation (*Figure 1b*, inset) at the neutral insert and the lengthy traversal times for dsMeP inserts of 5 bp or longer (*Figure 2c*) suggest that, given a sufficient number of traversal attempts, SpoIIIE can cross even relatively long stretches of dsMeP (10–30 bp). The drop in traversal probability for longer dsMeP inserts suggests that the motor can hold onto MeP DNA for a short amount of time during which it can either step forward, or lose its grip on the neutral DNA. In other words, forward translocation along the MeP insert is in kinetic competition with backward slipping (*Aathavan et al., 2009*). As a result, to traverse longer MeP inserts, the motor must execute a correspondingly larger number of consecutive productive power-strokes. We hypothesize that other types of motor–DNA interactions (e.g. steric) enable the motor to traverse the neutral insert given an arbitrarily large number of crossing attempts. In support of this idea, the φ29 ring ATPase has been shown to require electrostatic contacts with the DNA every 10 phosphates, but relies on steric, non-specific interactions to exert force and translocate the DNA in-between those electrostatic contacts (*Aathavan et al., 2009*).

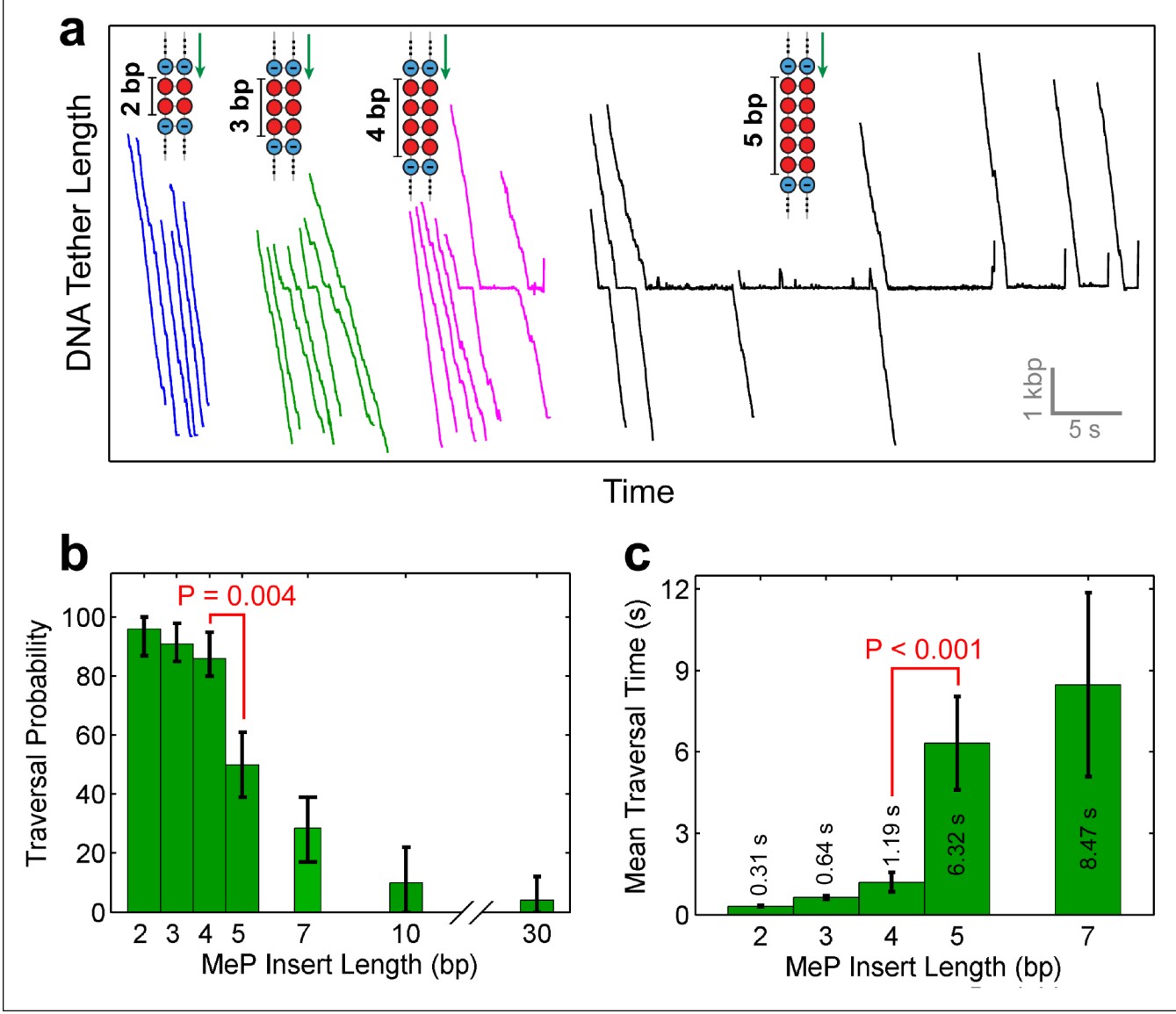

**Figure 2.** Monitoring SpoIIIE crossing of contiguous MeP inserts. (a) Sample traces of SpoIIIE translocating on DNA with dsMeP inserts of 2 bp (blue), 3 bp (green), 4 bp (magenta), and 5 bp (black). (b) Traversal probability for dsMeP inserts of various length. Note the sharp decrease in traversal probability between 4-bp and 5-bp inserts. Error-bars represent the 68% CI estimated via bootstrapping. (c) Mean traversal time for dsMeP inserts of various length. Note the large increase in traversal time between 4-bp and 5-bp inserts. Error-bars represent the SEM. P values calculated with a two-tailed Fisher exact test.

## SpoIIIE tracks the 5'→3' strand in the direction of translocation

To determine whether SpoIIIE tracks the phosphate backbone of one or both DNA strands, we repeated the above experiment with 30-base inserts where phosphates on either strand were selectively neutralized. 23 out of 24 SpoIIIE molecules traversed the insert containing MeP on the 3'→5' strand in the direction of translocation (*Figure 3a*, orange traces), compared to only 6 out of 28 SpoIIIE molecules that crossed the insert with a neutral backbone on the 5'→3' strand (*Figure 3a*, magenta traces). Strikingly, the traversals of the insert with a neutral 3'→5' strand were very fast (similar to translocation along unmodified DNA), whereas the traversals of the insert with a neutralized 5'→3' strand required 5–50 s, likely involving several crossing attempts. These results clearly indicate that SpoIIIE makes essential electrostatic contacts selectively with the charged backbone on the 5'→3' strand in the direction of translocation.

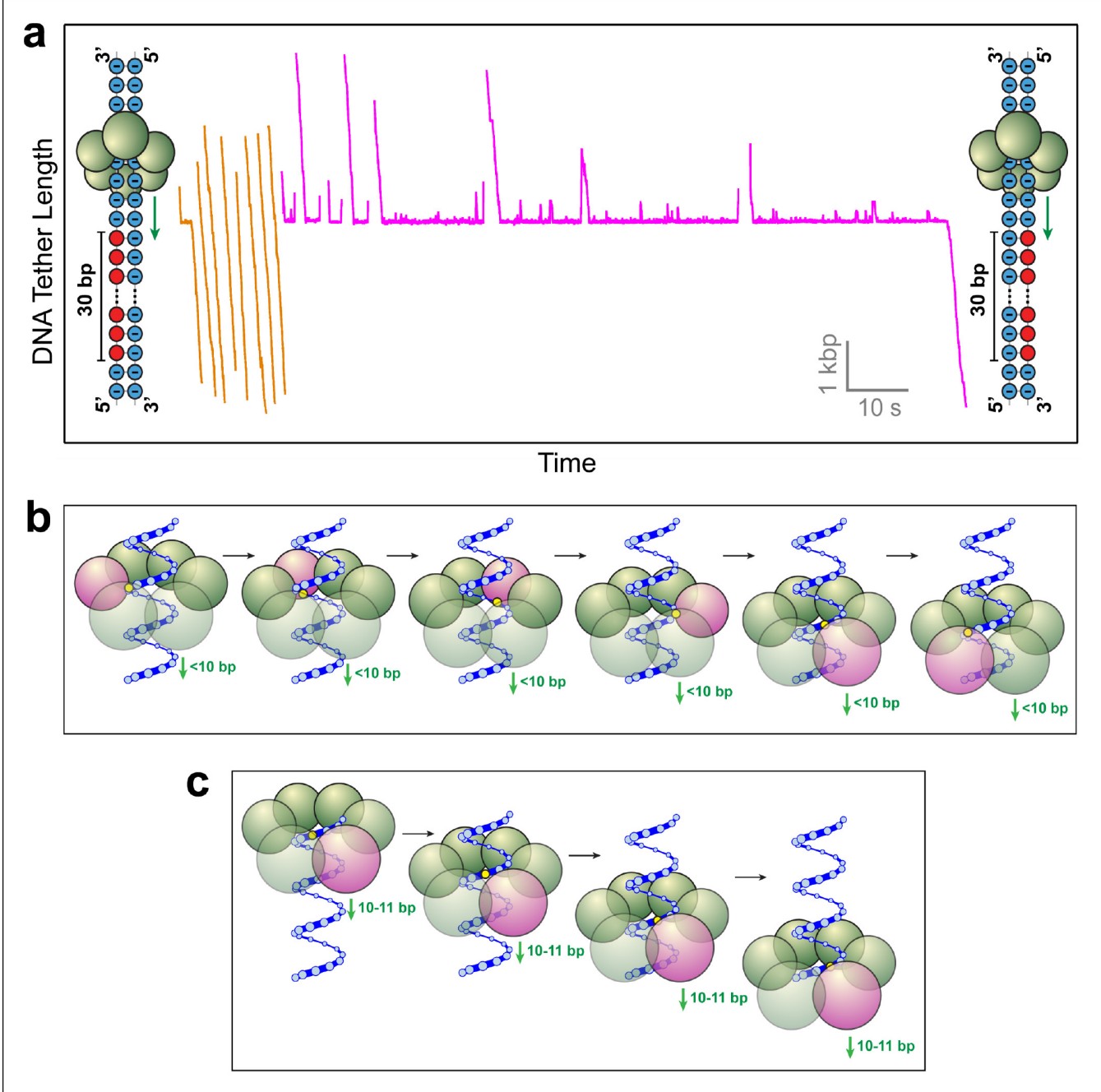

**Figure 3.** SpoIIIE favors phosphate interactions on the 5'–3' strand in the direction of translocation. (a) Sample traces of SpoIIIE translocating on DNA with 30 bp of neutral DNA on either the 3'–5' strand (orange traces) or the 5'–3' strand (magenta traces) in the direction of translocation. (b) Cartoon illustrating a hypothetical sequential model in which the DNA backbone is handed off between adjacent subunits within a hexameric ring. For clarity only one strand of dsDNA is shown. A highlighted backbone phosphate (yellow) is in contact with a motor subunit (magenta). In general, the motor step size in such a model can be anything less than 10 bp. Here we illustrate the model using a 2-bp step size. After one motor subunit (magenta) executes its power-stroke, the helical backbone of the dsDNA's will be shifted, positioning the phosphate backbone in close proximity to the next subunit poised to fire. (c) Cartoon illustrating a hypothetical model in which a 10-bp burst enables a hexameric ring ATPase to maintain phosphate contacts on the same DNA strand. Initially, a single subunit (magenta) is contacting the phosphate backbone (yellow). After a 10 bp burst, the motor has traversed nearly a full helical turn on dsDNA, bringing the phosphate backbone back in register with the same ATPase subunit.

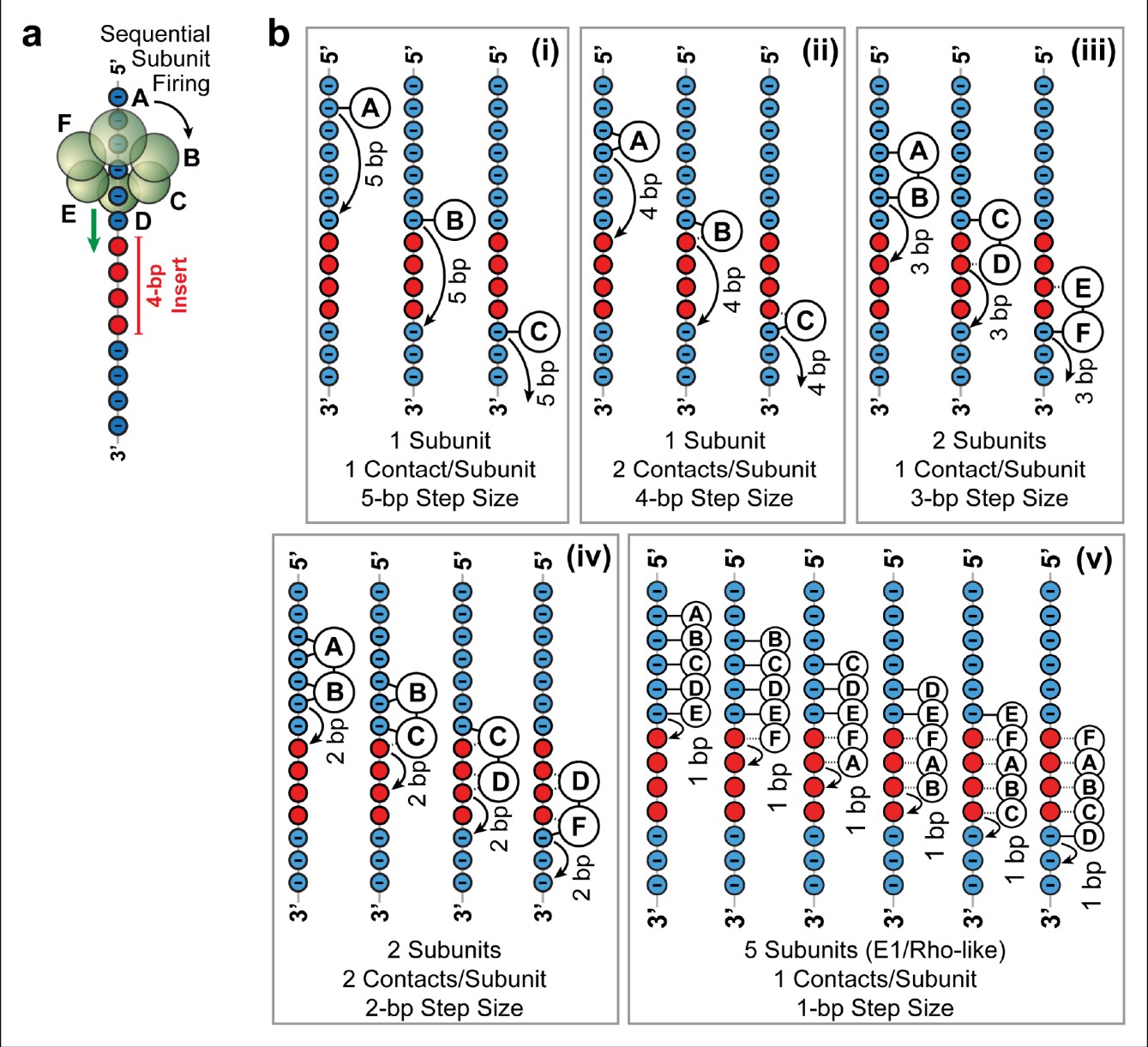

**Figure 4.** Possible SpoIIIE translocation models capable of crossing 4 bp of dsMeP. (a) Cartoon illustrating the sequential firing of subunits A–F of SpoIIIE as it approaches a 4 bp insert of dsMeP. (b) Possible translocation models (i–v) depicting the interaction between SpoIIIE subunits (A–F) and DNA; in all models subunits fire in a sequential order (A→B→C→D→E→F→A etc). For clarity, we show only the backbone the DNA strand tracked by SpoIIIE. Phosphate and MeP groups are shown in blue and red, respectively. A solid line connecting a SpoIIIE subunit to the backbone represents a stable electrostatic interaction; a dashed line represents a disrupted interaction.

## Strand tracking favors a sequential ATP hydrolysis model for SpoIIIE

The observation that SpoIIIE tracks only one of the DNA strands imposes geometric constraints on how the DNA is handed off from subunit to subunit during translocation. Strand-tracking is inconsistent with a stochastic coordination mechanism in which the six ATPase subunits execute their power-strokes at random. In such a mechanism, the subunit poised to fire is unlikely to be aligned with the tracked strand and will therefore not engage the DNA substrate. Since the DNA is held under constant tension in our experiments, a stochastically coordinated ATPase ring is expected to slip frequently, contrary to our observations.

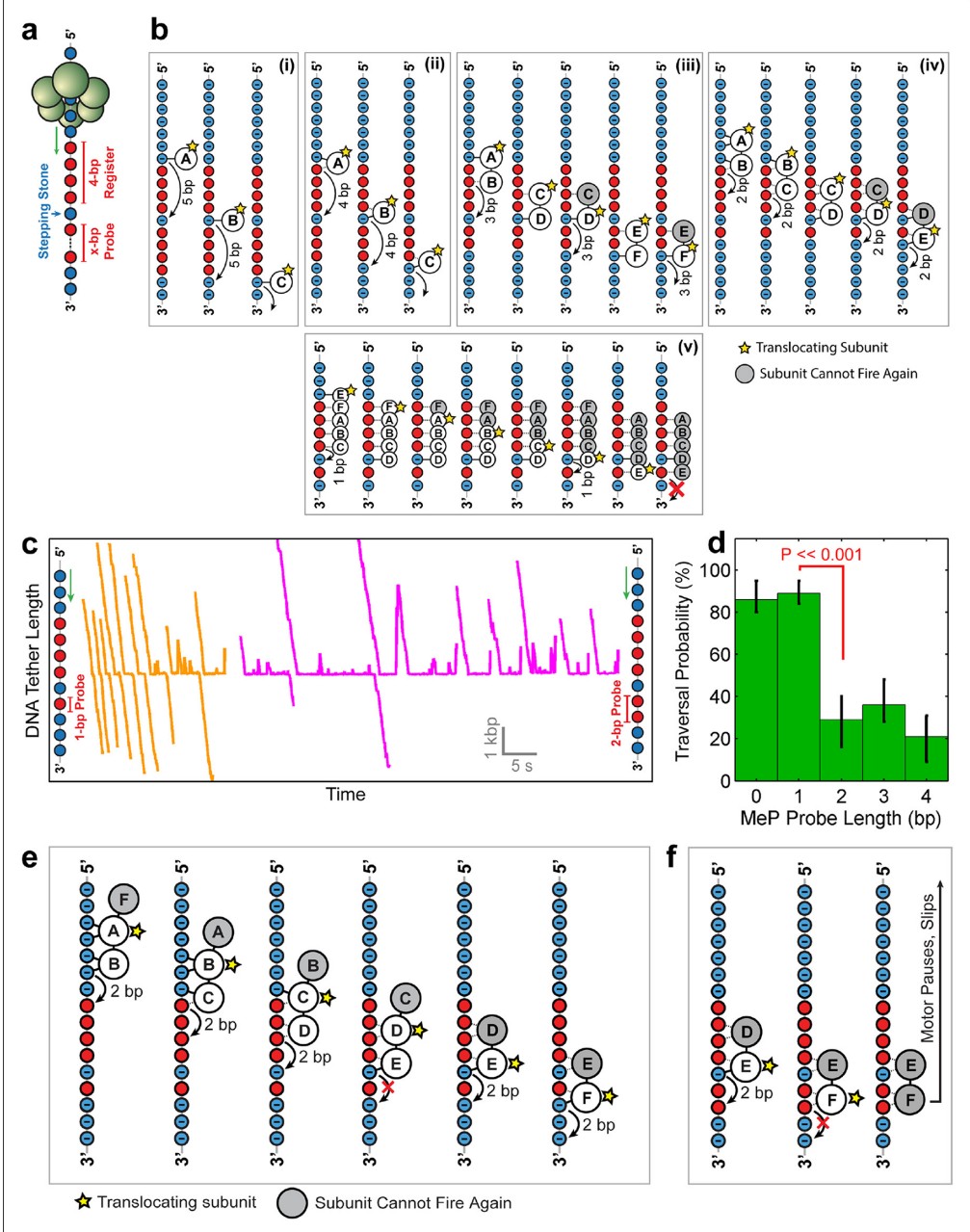

**Figure 5.** Deducing SpoIIIE's step size using 'stepping-stone' MeP constructs. (**a**) The design of 'stepping-stone' constructs. Each insert consists of a register segment with 4 bp of neutral DNA, followed by one regular DNA base ('stepping-stone'), and a probe segment with x bp of neutral DNA. For clarity, only the DNA strand tracked by SpoIIIE is shown. (**b**) Diagrams illustrating the longest probe segment that can be traversed by the models depicted in Figure 4b. Model (i) should traverse a probe segment of at most 4 bp, whereas model (v) cannot traverse even a probe of 1 bp. (**c**) Sample traces of SpoIIIE translocating on DNA with 'stepping-stone' inserts containing a 1-bp probe (orange) or a 2-bp probe (magenta). (**d**) Traversal probability for stepping-stone constructs with various probe lengths. Error-bars show the 68% CI estimated via bootstrapping. p values were calculated using a two-tailed Fisher exact test. (**e**) Diagram illustrating how model (iv) can successfully traverse a 'stepping-stone' insert with a probe of 1 bp. The star marks the subunit executing the power-stroke. Once a subunit fires, it cannot fire again (gray shading) and must eventually disengage from the DNA (subunit shown as making no interactions with DNA). (**f**) Diagram showing why model (iv) fails to cross a 'stepping-stone' insert with a probe of 2 bp.

*Figure 5. continued on next page*

*Figure 5. Continued*

The following figure supplements are available for Figure 5:

**Figure supplement 1.** Crossing predictions of the E1/Rho-like translocation models.

Two possible scenarios are consistent with strand–tracking: (i) Subunits hydrolyze ATP and execute their power-strokes sequentially around the ring, such that after every power-stroke the subunit scheduled to fire next is properly aligned to interact with the phosphate backbone of the tracked strand (*Figure 3b*). In this scenario, the SpoIIIE subunits would fire consecutively (subunit 1 fires, followed by subunit 2, followed by subunit 3, etc). After subunit 1 executes its power-stroke, the helical geometry of the DNA will position the phosphate backbone of the tracked strand in close proximity to subunit 2, depending on the step size of the motor (*Massey et al., 2006*; *Strick and Quessada-Vial, 2006*). Thus, motor-DNA contacts would proceed from one subunit to the next in an ordinal fashion. (ii) The motor translocates DNA in increments of 10–11 bp, which closely matches the helical periodicity of dsDNA (10.4–10.5 bp/pitch). This mechanism would ensure that after each translocation event, the motor contacts the same strand after traversing one full helical turn along the DNA (*Figure 3c*). Such a mechanism is employed by the φ29 ring ATPase, which translocates DNA in 10-bp bursts and contacts the DNA backbone on the same strand every 10 base pairs (*Aathavan et al., 2009*).

The data presented above indicate that SpoIIIE makes periodic contacts with the same DNA strand every five base pairs or less and are consistent only with the sequential ATP hydrolysis mechanism outlined in scenario (i).

## MeP stepping stone constructs suggest that SpoIIIE has a step size of 2 bp

In order to rationalize how SpoIIIE easily traverses 4 bp of neutral DNA, but has difficulty crossing 5 bp of neutral DNA (*Figure 2b*), we considered several motor-DNA contact models. To be consistent with the observation that SpoIIIE tracks only one DNA strand (*Figure 3a*), all potential models require that SpoIIIE subunits fire in a well-defined sequential order around the ring (*Figure 4a*). Furthermore, since electrostatic interactions with the backbone on one strand are critical for maintaining the motor's grip on DNA, we reason that at least one electrostatic contact between the motor and the phosphate backbone is needed for SpoIIIE to maintain a stable grip on its DNA substrate. Finally, the pool of potential models can be further narrowed by making an assumption about the state of dsDNA inside the central pore of SpoIIIE during translocation. Although no ring-shaped dsDNA translocase has been co-crystallized with its substrate, it is thought that dsDNA is not significantly distorted within the central pore of ring ATPases (*Massey et al., 2006*; *Sun et al., 2008*). For example, the central channel of the FtsK ring can fully accommodate a single, undistorted B-form dsDNA strand (*Massey et al., 2006*). Therefore, we considered models where the helical pitch of dsDNA inside the SpoIIIE channel is similar to that of B-form DNA (10.5 bp/turn). Expanding our models to include a distorted helix similar to A-form DNA (11.0 bp/turn) would require only minor corrections and would not affect the overall predictions of these models.

Five models (*Figure 4b i–v*) are consistent with the results of the dsMeP experiments (*Figure 2*) and satisfy the conditions listed above: (i) At any time, only one subunit contacts one phosphate on DNA (*Figure 4b i*); this model requires a 5-bp step size to cross a 4-bp MeP insert. (ii) Only one subunit contacts two adjacent DNA phosphates at any time (*Figure 4b ii*); this model requires a 4-bp step size to clear a 4-bp MeP insert. (iii) At any time, two neighboring subunits each contact one DNA phosphate; this model requires a 3-bp step size to clear a 4-bp MeP insert (*Figure 4b iii*). (iv) At any time, two neighboring subunits each contact two adjacent DNA phosphates; this model requires a 2-bp step size to cross a 4-bp MeP insert (*Figure 4b iv*). We disfavor models where SpoIIIE subunit simultaneously contact more than two consecutive phosphate groups on the DNA backbone, and models where three or more consecutive subunits contact the DNA backbone because such models would require large motor/DNA distortions. Finally, we also considered a translocation model similar to the mechanisms proposed for the E1 and Rho helicases (*Enemark and Joshua-Tor, 2006*; *Thomsen and Berger, 2009*), (v) at any time five subunits contact five consecutive DNA phosphates; this model requires a 1-bp step size to cross a 4-bp insert (*Figure 4b v*).

Although this E1/Rho-like model requires significant motor/DNA distortions (5 SpoIIIE subunits span an arc of 300° while five consecutive phosphates in dsDNA span an arc of ~170°), it is in principle consistent with the data from *Figure 2*.

While we cannot rule out more complex models that may involve non-consecutive subunits simultaneously contacting the DNA, or stochastic bursts consisting of multiple, rapid consecutive steps (*Cordova et al., 2014*; *Sen et al., 2013*), we favor the more parsimonious models presented here.

To distinguish among the models proposed above and to determine SpoIIIE's step size, we challenged the motor with inserts containing a 4-bp MeP segment ('register'), followed by 1 bp of regular DNA ('stepping-stone') that is in turn followed by a variable-length MeP segment ('probe') of 1–4 bp (*Figure 5a*). Because SpoIIIE easily crosses up to 4 bp of neutral DNA, the 4-bp register is meant to align the stepping phase of SpoIIIE with the stepping-stone segment. Motors that are not in perfect register with the stepping-stone should not establish electrostatic contacts with DNA and will either slip backward or stall. The motors that successfully traverse the entire insert should land on the stepping-stone in a 'front-foot-first' configuration—wherein only the leading motor-DNA contact is securely established—and should be able to cross a probe segment of length equal to the step size minus 1 bp (*Figure 5b*). We found that SpoIIIE traversed a 1-bp neutral probe with near-100% probability (*Figure 5c* orange traces), inconsistent with the 5 subunit contact model (*Figure 5b v* ). The crossing probability sharply drops to ~25% for 2-bp, 3-bp, or 4-bp neutral probes (*Figure 5c*, magenta traces and *Figure 5d*). As seen in *Figure 5b*, only model iv can easily cross a 1-bp probe but not probes of 2 bp or longer. In this model, two consecutive SpoIIIE subunits each contact two adjacent phosphates, and DNA is translocated in 2-bp steps (*Figure 5b iv*). A 2-bp step size is in good agreement with both enzymatic and structural studies of the related dsDNA translocase FtsK, which translocates 1.6–2.1 bp per ATP hydrolyzed (*Graham et al., 2010*; *Massey et al., 2006*).

*Figure 5e* illustrates how model iv enables SpoIIIE to cross the insert with a 1-bp probe. The first three frames illustrate the sequential firing of subunits A, B, and C, all of which maintain at least one anchoring contact with the phosphate backbone. Because subunit D cannot make electrostatic contacts with the backbone, it fails to propel the DNA upon firing (*Figure 5e*, frame 5). As a result, no other subunit can establish new contacts with the DNA backbone. After subunit E translocates the DNA by 2 bp, subunit F can now latch onto a negatively charged phosphate and continue to translocate DNA. *Figure 5f* illustrates how this model copes with a 2-bp MeP probe: subunit F cannot anchor itself onto the DNA, causing the motor to pause and eventually slip (as shown in *Figure 5c*, magenta traces). This two-subunit DNA escort model requires one subunit to execute the power stroke while an adjacent subunit maintains its phosphate contacts during this power-stroke, escorting the DNA through the ring. In this model, the ATPase subunits translocate the DNA sequentially around the ring in a highly coordinated fashion that enables SpoIIIE to track one DNA strand. There is increasing structural evidence for this type of motor mechanism involving 'translocating' subunits and 'escorting' subunits; examples include the *E.coli*'s Rho helicase (*Thomsen and Berger, 2009*) and the papillomavirus E1 helicase (*Enemark and Joshua-Tor, 2006*). However, unlike the E1/Rho mechanisms, which employ four escorting subunits, the SpoIIIE translocation model proposed here requires only one escorting subunit.

## Nucleotides stabilize the SpoIIIE-DNA interactions

To quantify the strength of the motor–DNA interaction, we measured the force at which SpoIIIE loses its grip on DNA. In a buffer lacking nucleotides, SpoIIIE could bind to DNA and form tethers between the trapped bead and the micropipette-held bead. Manual pulling experiments revealed that these tethers rupture at ~3 pN (*Figure 7—figure supplement 1*, apo), suggesting that apo-SpoIIIE does not interact strongly with DNA. In buffers containing only ADP or ATPγS, SpoIIIE could bind to DNA, forming tethers that rupture at ~15 and ~25 pN, respectively (*Figure 7—figure supplement 1*). These results indicate that nucleotides stabilize motor–DNA interactions and suggest that only nucleotide-bound subunits are capable of forming stable electrostatic contacts with the DNA phosphate backbone.

## The SpoIIIE ring can operate with non-consecutive inactive subunits

To further investigate how individual SpoIIIE subunits coordinate their ATP hydrolysis activity, we monitored DNA translocation in a saturating [ATP] buffer that contained ATPγS – a nucleotide

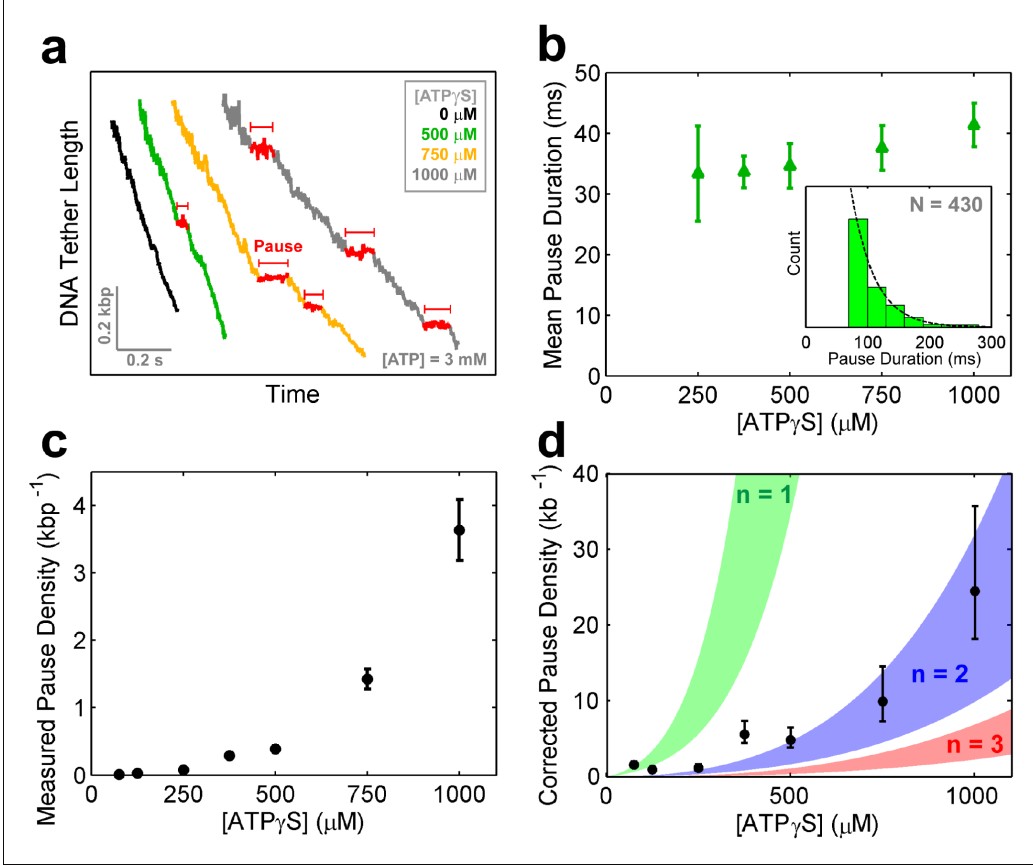

**Figure 6.** SpoIIIE pauses when two consecutive subunits are bound to ATPγS. (a) Sample SpoIIIE traces acquired at different [ATPγS] and 3mM ATP. Pauses are highlighted in red. (b) Mean duration of the ATPγS-induced pauses extracted from fitting the pause duration distribution to an exponential. Error-bars represent the 95% CI of the fit. Insert: histogram of pause durations at 750 μM ATPγS and 3 mM ATP and the exponential fit to this distribution. (c) The density of ATPγS-induced pauses at various [ATPγS] and 3mM ATP. Error-bars represent the SEM. (d) Pause density versus [ATPγS] corrected to account for missed pauses (black symbols). Error-bars represent the SEM. The shaded regions illustrate the predictions for a sequential hydrolysis model where SpoIIIE pauses when n consecutive subunits are bound to ATPγS.

The following figure supplements are available for Figure 6:

**Figure supplement 1.** SpoIIIE activity inhibition with ATP analogs.

analog that is hydrolyzed very slowly. Structural studies of FtsK indicate that ATPγS binds to the same catalytic pocket as ATP (*Massey et al., 2006*), suggesting that ATPγS should act as a competitive inhibitor to ATP binding for the SpoIIIE/FtsK family of ring ATPases. We observe that ATPγS induces SpoIIIE pausing (*Figure 6a*) and reduces the pause-free translocation velocity (*Figure 6—figure supplement 1a*). This reduction in pause-free velocity is well described by a simple competitive inhibition mechanism (*Figure 6—figure supplement 1b*). Similar results were obtained with a non-hydrolyzable ATP analog–AMP-PNP (*Figure 6—figure supplement 1c–d*). Taken together, these results indicate that nucleotide analogs induce SpoIIIE pausing by binding to a subset of ring subunits and arresting them in a pre-hydrolysis state.

To determine how many analog-bound subunits are required to arrest DNA translocation, we measured the density and duration of ATPγS-induced pauses as a function of [ATPγS]. The density and duration of analog-induced pauses does not depend on external force (*Figure 6—figure supplement 1e–f*), consequently pauses detected at all forces were pooled together. Given the spatio-temporal resolution of our experiments, we could reliably detect pauses longer than 30 ms at 125 – 250 μM ATPγS, 50 ms at 375–500 μM ATPγS, and 75 ms at 750–1000 μM ATPγS. However,

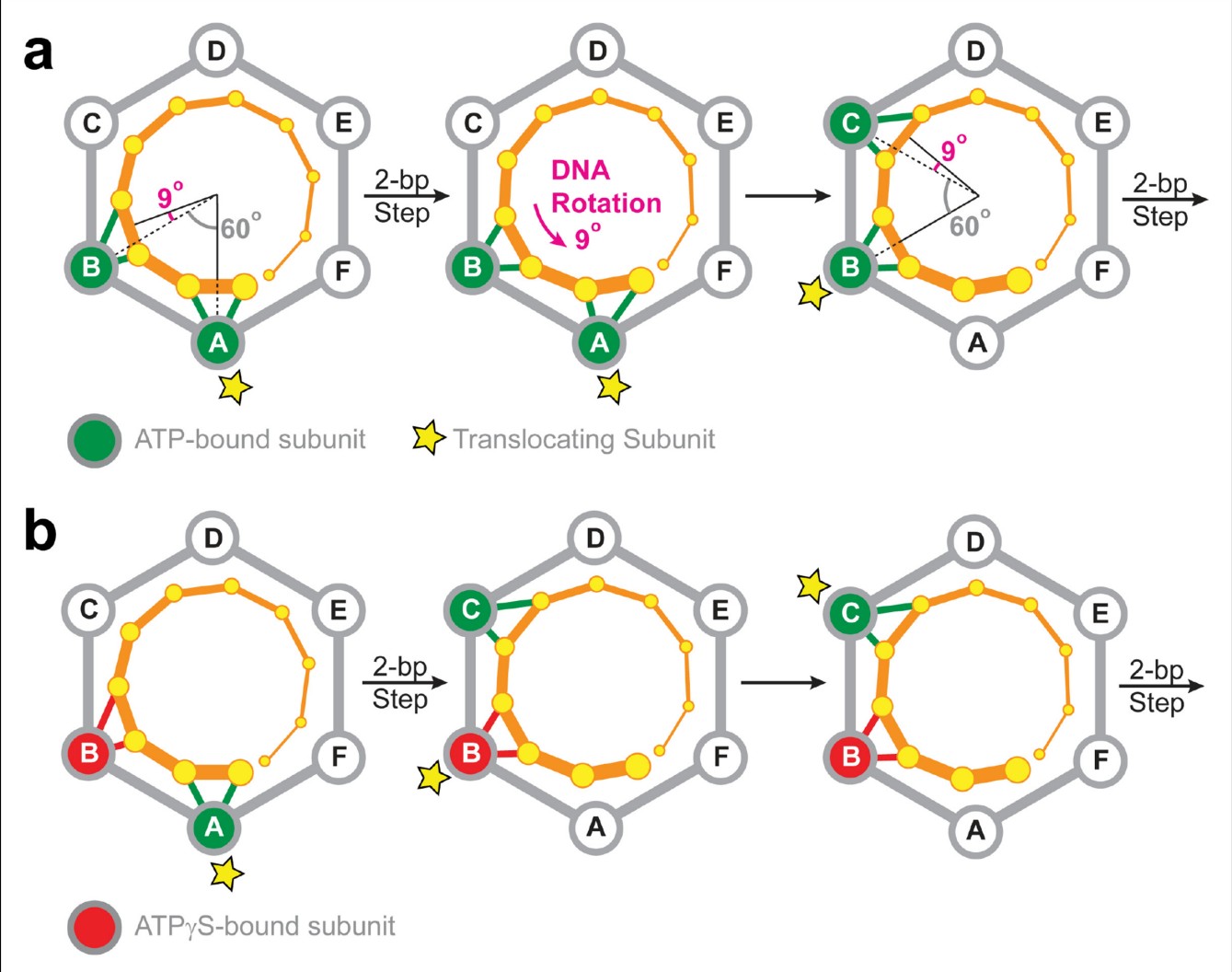

**Figure 7.** Two-subunit DNA escort model. (a) Diagram illustrating how SpoIIIE supercoils DNA while tracking the phosphate backbone of one DNA strand. Neighboring subunits in the SpoIIIE hexamer are spaced by 60°; consecutive DNA phosphates are spaced by ~34°; and the backbone contacts of two adjacent subunits are separated by ~69°. After a 2-bp translocation step, a 9-degree counter-clockwise rotation of the DNA relative to the motor is needed to align the next translocating subunit and the nearest pair of backbone phosphates. (b) Diagram of the SpoIIIE hexamer illustrating how the two-subunit DNA escort model enables the motor to bypass an ATPγS-bound subunit (red). For simplicity, only one DNA strand is shown (orange). DNA is translocated out of the page, and each subunit interacts with two neighboring phosphates. The star marks the subunit slated to execute the power-stroke.

The following figure supplements are available for Figure 7:

**Figure supplement 1.** Assessing the strength of the SpoIIIE–DNA interaction in the presence of different nucleotides.

regardless of the shortest detectable pause duration and the ATPγS concentration, the distribution of measured pause durations was well described by a single exponential decay (*Figure 6b*, inset) with the same characteristic life-time of ~35 ms (*Figure 6b*). The fact that the characteristic life-time does not depend on ATPγS concentration suggests that all analog-induced pauses are drawn from the same distribution. Furthermore, the single-exponential dependence of the pause duration distribution indicates that the exit from the paused state is governed by a single rate-limiting event—presumably the exchange of an ATPγS molecule with one ATP which is present in saturating concentrations, as proposed for other ring ATPases (*Chistol et al., 2012*; *Sen et al., 2013*). Despite the fact that we cannot detect analog-induced pauses shorter than a certain cutoff (30–75 ms), we can estimate the number of missed pauses—and therefore the true pause density—from the

measured pause density (*Figure 6c*) and the characteristic life-time derived from the pause distribution (*Figure 6b*), assuming that the single-exponential distribution holds for pauses shorter than the cutoff (*Hodges et al., 2009*) (see 'Materials and methods').

The MeP experiments support a sequential nucleotide hydrolysis model. Therefore, to explain the density of ATPγS-induced pauses, we considered a sequentially coordinated hexameric ATPase ring that pauses whenever *n* consecutive subunits are bound to ATPγS ($1 \leq n \leq 6$). The pause density (*PD*) can be analytically expressed in terms of the motor's pause-free velocity ($v_{pf}$), the ATPγS dissociation rate ($k_{off}$), and the probability that the motor is paused ($P_{pause}$)—which is proportional to the concentrations and dissociation constants of ATP and ATPγS (Materials and methods), as follows:

$$PD = \frac{k_{off}}{v_{pf}} \cdot \frac{P_{pause}}{(1 - P_{Pause})}$$

*Figure 6d* shows the expected pause density predicted for different *n* values. For example, *n*=1 (green shaded region) corresponds to a sequentially coordinated ring that pauses whenever a single subunit is bound to ATPγS; *n*=2 (blue shaded region) corresponds to a sequentially coordinated ring that pauses when two consecutive subunits are both bound to ATPγS, and so on. The ATPγS pause-density data corrected to account for missed pauses (*Figure 6d* black symbols) are best described by the model in which two (n=*2*) consecutive analog-bound subunits are required to induce a pause in the motor (*Figure 6d*, blue-shaded region). In other words, analysis of the pause density indicates that the motor can operate processively with non-consecutive inactive subunits but not with two or more consecutive subunits. This conclusion is further supported by the observation that FtsK hexamers readily bypass individual catalytically inactive ATPase subunits (*Crozat et al., 2010*).

## Discussion

### Secondary motor-DNA interactions

We have shown that SpoIIIE translocates DNA by making crucial anchoring contacts with the negatively charged phosphate groups on one DNA strand. Although electrostatic interactions with the backbone of the 5'→3' strand in the direction of translocation are the principal mode of motor-substrate contact, we surmise that other types of interactions (e.g. steric) play a secondary role in maintaining SpoIIIE's grip on DNA. This inference explains why we observe small but non-zero traversal probabilities for neutral inserts of 10–30 bp, much longer than the expected motor step size (*Figure 2b*). Interestingly, SpoIIIE is more likely to traverse 30-bp inserts with a neutral backbone only on the 5'→3' strand (6 out of 28 molecules) than 30-bp inserts with a neutral backbone on both strands (1 out of 32 molecules) (*Table 1*). To explain this discrepancy, we speculate that during translocation, in addition to the essential electrostatic interactions with the 5'→3' strand, the SpoIIIE ring may also interact very weakly with charges on the 3'→5' strand. These secondary interactions could be mediated by parts of the motor other than the pore loops which execute the power – stroke.

### MeP is not expected to cause significant DNA distortions

Prior experiments on MeP DNA have established that certain patterns of neutral and charged bases can introduce large DNA distortions. An asymmetric neutralization of the phosphate backbone whereby one 'face' of the dsDNA helix had its charges removed (e.g. neutralizing bases 1-3 on one strand and bases 3-6 on the other strand) results in DNA bending toward the direction of the neutralized region (*Strauss and Maher, 1994*). However, a symmetric neutralization of the phosphate backbone, whereby the phosphates on both strands were uniformly neutralized, resulted in no significant DNA distortions (*Strauss and Maher, 1994*).

We do not expect the MeP DNA constructs used in this study to introduce significant distortions on the double helix for two reasons: (1) Magnesium ions were present in the experimental buffer at a sufficiently high concentration (10 mM), which had been demonstrated to mitigate the effect of DNA distortions due to asymmetric phosphate neutralization (*Strauss and Maher, 1994*). (2) In designing the MeP inserts used in this study, care was taken to avoid an asymmetric neutralization of the phosphate backbone. The majority of the experiments with MeP inserts tested in this study utilized dsMeP modifications. Thus, the backbone neutralization was symmetrically distributed and therefore not expected to cause significant DNA distortions. We tested only two single-stranded

MeP/DNA hybrid constructs, with 30 consecutive neutralized bases on either strand, covering nearly three full helical turns of the DNA. Because the backbone neutralization was distributed evenly in all directions around the DNA helix, the charge neutralization should be symmetrical, and therefore, it is not expected to introduce significant DNA distortions.

## DNA translocation model

To rationalize the results of the MeP experiments, we propose a minimal translocation model in which SpoIIIE subunits fire sequentially around the ring, each subunit translocates 2 bp of DNA per power-stroke, at least two consecutive ATPase subunits contact the backbone of one DNA strand, and each subunit contacts two consecutive backbone phosphates. To be consistent with the results of the stepping-stone insert experiments (*Figure 5c–d*), our two-subunit translocation-escort model requires that subunits fire in a well- defined sequential fashion (A, B, C, D, E, F, A, B, C etc) and a subunit cannot fire multiple times in a row or out of order.

## SpoIIIE does not employ the E1/Rho-like translocation model

The structures of homo-hexameric helicases E1 and Rho co-crystallized with their single-stranded nucleic acid substrates strongly support a translocation mechanism where five motor subunits contact five consecutive phosphates on the ssDNA/ssRNA backbone (*Enemark and Joshua-Tor, 2006*; *Thomsen and Berger, 2009*), and the hydrolysis of one ATP is coupled to the translocation of one nucleotide. Although an E1/Rho-like mechanism could potentially rationalize how SpoIIIE easily traverses a dsMeP insert of 4 bp but not 5 bp (*Figure 2*), such a model predicts that the motor would fail to traverse a stepping-stone construct with a 1-bp MeP probe (*Figure 5b v*). Even if the E1/Rho-like model did allow a subunit to fire several times in a row, such a model would still be inconsistent with the stepping stone data (*Figure 5—figure supplement 1*).

## Strand-tracking in a dsDNA translocase

It was previously reported that SpoIIIE supercoils plasmid DNA *in vitro*, and strand/groove-tracking was proposed to explain this observation (*Bath et al., 2000*). Magnetic tweezers measurements of DNA supercoiling during translocation ruled out a groove-tracking mechanism for FtsK (*Saleh et al., 2005*). Here, we show that SpoIIIE does indeed track one DNA strand by making specific electrostatic contacts with backbone phosphates on the 5′→3′ strand in the direction of translocation, a feature reported for other RecA-like NTPases: the φ29 packaging motor, DnaB, and Rho (*Aathavan et al., 2009*; *Itsathitphaisarn et al., 2012*; *Thomsen and Berger, 2009*). Unlike the φ29 ATPase, which translocates dsDNA, DnaB, and Rho are single-stranded nucleic acid translocases, and therefore track one strand by default. Interestingly, both SpoIIIE and the φ29 ATPase encircle dsDNA, positioning multiple subunits in close proximity to the dsDNA helix, while they still possess the ability to discriminate one strand from the other. How these dsDNA translocases achieve strand discrimination remains unclear.

Several studies have demonstrated that certain skewed sequences (i.e. sequences whose presence is biased on the leading vs. the lagging strand) impart directionality to FtsK/SpoIIIE translocation (*Besprozvannaya et al., 2013*; *Cattoni et al., 2013*; *Lee et al., 2012*; *Levy et al., 2005*; *Löwe et al., 2008*). It is tempting to imagine that the strand tracking mechanism presented here could be used by SpoIIIE to read these skewed sequences (i.e. the SpoIIIE Recognition Sequences, SRS). However, the γ domains of SpoIIIE have been shown to be required for reading the SRS (*Ptacin et al., 2008*) and co-crystal structures of dsDNA with the domain γ reveals it interacting with both DNA strands at the backbone, the major and minor grooves, and individual bases (*Löwe et al., 2008*). We consider it unlikely therefore, that SpoIIIE enlists the strand-tracking mechanism described here for SRS sequence recognition.

## The motor-DNA symmetry mismatch should give rise to DNA supercoiling

The electrostatic interactions between SpoIIIE and the DNA backbone are likely to have two main purposes *in vivo*: (i) They serve a load-bearing function by providing SpoIIIE with the grip necessary to strip-bound proteins off the DNA at the high speeds at which the motor operates (*Marquis et al., 2008*); and (ii) They enable the motor to supercoil the DNA during translocation. The later arises

from the symmetry mismatch between the angles spanned by two adjacent SpoIIIE subunits and the angle that separates the phosphates contacted by the motor during a 2-bp step. *Figure 7a* depicts the SpoIIIE-DNA complex as seen from the N terminus of the motor; in this view, SpoIIIE translocates DNA toward the viewer (*Massey et al., 2006*). In the two-subunit DNA escort model, the DNA is handed over from one subunit to the next and, for each 2-bp step, the motor-DNA contacts rotate clockwise around the ring by 60°. However, since B-form DNA has a helical pitch of 10.4 bp/turn (*Wang, 1979*), the dsDNA backbone rotates clockwise by 69° for every 2 bp. Therefore, after each motor step, a counter-clockwise 9° rotation of the DNA relative to the motor is required to align the phosphate backbone with the next translocating subunit. As a result, the two-subunit escort model predicts that SpoIIIE should introduce one supercoil for every ~80 bp of DNA translocated. The magnitude and direction of this prediction is consistent with both *in vivo* measurements (*Nicholson and Setlow, 1990*) and *in vitro* single-molecule experiments. Indeed, magnetic tweezers experiments revealed that SpoIIIE supercoils DNA by one turn per 100 ± 10 bp of DNA translocated (Jerod Ptacin and Marcelo Nollman, personal communication). It was previously reported that strand tracking by the φ29 ATPase, another ring-shaped dsDNA motor, also leads to DNA supercoiling (*Liu et al., 2014*). The results presented here for SpoIIIE suggest that DNA supercoiling may be a general feature of ring-shaped dsDNA motors that track one strand. The amount and direction of supercoiling should depend on the (i) the helical pitch of dsDNA, (ii) the periodicity of motor-DNA contacts (step size), and (iii) the motor's inter-subunit coordination mechanism.

*In vivo* measurements indicate that, during *B. subtilis* sporulation, the DNA is negatively supercoiled by 1 turn per 97 ± 6 bp in the forespore and 1 turn per 140 ± 10 bp in the mother cell (*Nicholson and Setlow, 1990*). Since SpoIIIE is anchored at the septum during sporulation and the *B. subtilis* DNA is circular, the two-subunit escort model predicts that SpoIIIE translocation should introduce one negative supercoil in the forespore and one positive supercoil in the mother cell for every ~80 bp translocated. We surmise that the SpoIIIE mechanism is fine tuned to deliver DNA to the forespore in the appropriate negative supercoiled state, similar to what has been proposed for FtsK (*Saleh et al., 2005*). This mechanism may help to conserve cellular resources by minimizing the amount of maintenance performed by topoisomerases/gyrases, a strategy that may provide a significant advantage in the harsh conditions that prompt sporulation (i.e., starvation).

## SpoIIIE/FtsK rings can tolerate non-consecutive inactive subunits

The results of ATPγS experiments suggest that SpoIIIE can processively translocate DNA with non-consecutive inactive subunits. This conclusion can be rationalized by the two-subunit DNA escort model proposed above: having two adjacent ATPase subunits simultaneously contact the DNA enables the motor to continue translocating if either of the two subunits is inactive, but not when both are disabled. *Figure 7b* illustrates how the two-subunit DNA escort model can bypass the ATPγS-bound subunit B: (i) subunit A fires and translocates DNA while subunit B escorts the DNA, (ii) the analog-bound subunit B fails to fire while subunit C was poised to escort the DNA (iii) subunit C then fires and translocates the DNA, handing it over to subunit D. During this step, subunit B, which remains bound to ATPγS, escorts the DNA for subunit C.

Our conclusions agree with the results of a single-molecule study by Crozat et al., which found that FtsK hexamers with two diametrically opposed inactive subunits translocated DNA as fast as wildtype hexamers (*Crozat et al., 2010*). The authors of that study proposed a sequential DNA escort mechanism in which at least three motor subunits contact the DNA at any given time. The ATPγS experiments presented here provide additional evidence that SpoIIIE/FtsK motors can bypass individual inactive subunits. Moreover, the strand-tracking behavior reported here for SpoIIIE strongly favors a model in which ring subunits fire sequentially, a key aspect of the DNA escort mechanism proposed here and in the Crozat et al study.

This study presents evidence for a type of inter-subunit coordination in ASCE ring NTPases where subunit firing is highly coordinated around the ring, yet the motor possesses sufficient flexibility to bypass non-consecutive inactive subunits. How is this mechanism optimized for the specific biological task of SpoIIIE? *In vivo*, SpoIIIE is present in low copy numbers during sporulation, with only two motors responsible for the vital task of chromosome translocation at any given time (*Burton et al., 2007*; *Yen Shin et al., 2015*). As a result, each SpoIIIE ring functions as a single-molecule bottleneck for this process, where the failure of either motor is likely to be lethal for the cell. The coordination

mechanism proposed here can potentially explain how *B. subtilis* safeguards against the failure of individual motor subunits during sporulation. We speculate that other ASCE motors that also represent single-points of failure may have evolved into similar flexible operations.

## Materials and methods

### Sample preparation

Biotinylated SpoIIIE constructs were generated by ligating the biotin tag sequence from plasmid Pinpoint Xa-1 (Promega, Madison, WI) to the N terminus of SpoIIIE from plasmid pJB103 (*Bath et al., 2000*). Protein purification was conducted as described previously with the addition of 2 μM biotin in the liquid cultures (*Ptacin et al., 2008*). DNA tethers were generated using a 5′ biotinylated primer (IDT) to PCR amplify a 21 kb region of lambda phage DNA (NEB) and gel extracted. DNA oligos containing MeP inserts (Gene Link or TriLink BioTechnologies) were ligated to gel-purified DNA fragments: a 9167-bp fragment with the ACT 3′-overhang (amplified from λ DNA and digested with AlwNI), and a biotinylated 3976-bp fragment with the GAA 3′-overhang (amplified from φ29 DNA and digested with BglI). The final ligation product was gel-purified using the QIAEX II kit (Qiagen).

### Optical tweezers experiments

In this study, 2.1 μm streptavidin beads (Spherotech) were blocked for 30 min in 50 mM Tris–HCl pH 7.5, 10 mM $MgCl_2$, 4% bovine serum albumin (BSA), w/v and 0.1% Tween-20. Then ~1 pmol of biotinylated SpoIIIE and ~1 ng of biotinylated DNA were incubated separately onto streptavidin beads and spatially separated in the fluidics chamber. DNA-bound beads and SpoIIIE-bound beads were brought in close proximity to allow SpoIIIE to engage the DNA. DNA translocation was conducted in a reaction buffer containing 50 mM Tris–HCl pH 7.5, 10 mM $MgCl_2$, and 3 mM ATP.

### Data analysis

Pauses were detected using a modified Schwartz Information Criterion (mSIC) method (*Chistol et al., 2012*). The number and duration of pauses missed by this algorithm were inferred by fitting the pause durations to a single exponential with a maximum likelihood estimator. After removing the detected pauses, the translocation velocity was computed by fitting the data to a straight line. Force-velocity data (*Figure 6—figure supplement 1a*) was gathered in 'passive-mode' (*Smith et al., 2001*), where the optical trap position is fixed. Single-molecule trajectories were partitioned into segments spanning 2–3 pN, and the velocity was computed for each segment. Tether tension and extension were converted to contour length using the Worm-Like-Chain approximation with persistence length p=30 nm, and stretch modulus S = 1200 pN·nm (*Baumann et al., 1997*).

### Estimating the number and duration of missed pauses

The pause-detection algorithm can detect only pauses longer than $t_c$, but it is possible to account for the duration and number of pauses missed by the algorithm. We observe that pauses are drawn from a single-exponential distribution *P(t)* with a mean pause-duration t.

$$P(t) = A \cdot e^{-\frac{t}{\tau}} \tag{1}$$

The number of all pauses ($N_{all}$), the number of detected pauses ($N_{det}$), and the number of missed pauses ($N_{miss}$) are given below.

$$N_{all} = \int_0^\infty P(t) \cdot dt = \int_0^\infty A \cdot e^{-\frac{t}{\tau}} \cdot dt \tag{2}$$

$$N_{det} = \int_{t_c}^\infty A \cdot e^{-\frac{t}{\tau}} \cdot dt \tag{3}$$

$$N_{miss} = \int_0^{t_c} A \cdot e^{-\frac{t}{\tau}} \cdot dt \tag{4}$$

The duration of all pauses ($T_{all}$)

$$T_{all} = \int_0^\infty t \cdot P(t) \cdot dt = \int_0^\infty t \cdot A \cdot e^{-\frac{t}{\tau}} \cdot dt \qquad (5)$$

The duration of detected pauses is $T_{det}$:

$$T_{det} = \int_{t_c}^\infty t \cdot A \cdot e^{-\frac{t}{\tau}} \cdot dt \qquad (6)$$

The duration of missed pauses:

$$T_{miss} = \int_0^{t_c} t \cdot A \cdot e^{-\frac{t}{\tau}} \cdot dt \qquad (7)$$

Given the number of detected pauses and the mean pause duration calculated by fitting the pause distribution to a single-exponential, it is possible to infer the number of total pauses and the number of missed pauses as follows:

$$N_{all} = N_{det} \cdot e^{\frac{t_c}{\tau}} \qquad (8)$$

$$N_{miss} = N_{det} \cdot \left( e^{\frac{t_c}{\tau}} - 1 \right) \qquad (9)$$

If the pause-detection algorithm identifies and removes only pauses longer than $t_c$, then the measured pause-free velocity ($V_{meas}$) is given by the following expression:

$$V_{meas} = \frac{D}{T_{pf}T_{miss}} \qquad (10)$$

Here $D$ is the total distance translocated by the motor, $T_{pf}$ is the 'pause-free' time (i.e. only the time spent translocating DNA), and $T_{miss}$ is the total duration of missed pauses. The true pause-free velocity ($V_{pf}$) is given by the following expression:

$$V_{pf} = \frac{D}{T_{pf}} = \frac{D}{\frac{D}{V_{meas}} - \int_0^{t_c} t \cdot A \cdot e^{-\frac{t}{\tau}} \cdot dt} \qquad (11)$$

## Predicting the density of ATPγS-induced pauses

The probability of finding the ring in a pause state $P_{pause}$ (i.e. the fraction of the time spent in the paused state) is:

$$P_{pause} = \frac{\tau_p}{\tau_p \tau_x} \qquad (12)$$

where $t_p$ is the total time the motor spends in a pause state and $t_x$ is the total time the motor spends translocating DNA. The total pause time $t_p$ and the total translocation time $t_x$ can be expressed as

$$\tau_p = <T> \cdot \eta = (1/k_{off}) \cdot \eta \qquad (13)$$

$$\tau_x = \frac{x(t)}{v_{PF}} \qquad (14)$$

where $<T>$ is the average pause duration, $\eta$ is the number of pauses, $k_{off}$ is a first-order dissociation rate of ATPγS from the ring, $x(t)$ is distance translocated over time and $v_{PF}$ is the pause-free velocity expressed as:

$$v_{PF} = \sum_{i=0}^6 p_i v_i \qquad (15)$$

where $p_i$ is the probability of the ring being bound to $i$ ATPγS molecules and $v_i$ is the pause-free velocity of the ring bound to $i$ ATPγS molecules. Substituting *Equation 14* into *Equation 12*, we can express the pausing probability as:

$$P_{pause} = \frac{\eta/x(t)}{\eta/x(t) + k_{off}/v_{PF}} = \frac{PD}{PD + k_{off}/v_{PF}} \tag{16}$$

We define the pause density (the number of pauses per distance translocated) as $PD = \eta / x (t)$. Thus, the PD can be expressed in terms of the pausing probability ($P_{pause}$), the ATPγS dissociation rate ($k_{off}$), and the pause-free translocation velocity ($v_{PF}$):

$$PD = \frac{k_{off}}{v_{PF}} \cdot \frac{P_{pause}}{(1 - P_{pause})} \tag{17}$$

The pausing probability ($P_{pause}$) can be expressed in terms of the probability that a single subunit is bound to ATPγS ($p$). $p$ depends on the concentrations of ATP and ATPγS, as well as the dissociation constants ($k_{off}/k_{on}$) of ATP and ATPγS ($K_{ATP}$ and $K_{\gamma S}$) for individual ATPase subunits (*Sen et al., 2013*):

$$p = \frac{K_{ATP}[ATP\gamma S]}{K_{\gamma S}K_{ATP}K_{\gamma S}[ATP] + K_{ATP}[ATP\gamma S]} \tag{18}$$

To calculate the expected pause probability, we used the measured $K_i$ of ATPγS (124 ± 20 μM) for $K_{\gamma S}$ (*Figure 6—figure supplement 1b*). For $K_{ATP}$, we used the measured $K_m$ of ATP (505 ± 50 μM) as an upper-bound estimate. For a sequential ordinal ATP binding/hydrolysis inter-subunit coordination model (i.e. subunit 1 binds ATP, hydrolyzes ATP, releases $P_i$ and ADP, and executes the power-stroke, followed by subunit 2, then subunit 3, etc), the pausing probability is given by $P_{pause} = p^n$, were $n$ is the number of ATPγS molecules required to induce a pause.

## Acknowledgements

We thank JR Moffitt, S Liu, M Sen, D Goldman, M Dangkulwanich, S Tafoya for critical reading of the manuscript; C Caffarro and L Alexander for help with preliminary experiments; J Ptacin and M Nollman for sharing unpublished data; SB Smith for instrument training and assistance; JY Shin, M Righini, B Onoa, and C Diaz for discussions. This work was supported by the Howard Hughes Medical Institute (to CB), by NIH grant R01GM071552 and R01GM032543, and by the US Department of Energy, Office of Basic Energy Sciences, Division of Materials Sciences and Engineering Nanomachine Program under Contract No. DE-AC02-05CH11231 (single molecule force measurement).

## Additional information

### Funding

| Funder | Grant reference number | Author |
|---|---|---|
| Howard Hughes Medical Institute | | Carlos Bustamante |
| National Institutes of Health | R01GM071552 | Carlos Bustamante |
| National Institutes of Health | R01GM032543 | Carlos Bustamante |
| U.S. Department of Energy | DE-AC02-05CH11231 | Carlos Bustamante |

The funders had no role in study design, data collection and interpretation, or the decision to submit the work for publication.

### Author contributions

NL, GC, Conception and design, Acquisition of data, Analysis and interpretation of data, Drafting or revising the article; CB, Conception and design, Drafting or revising the article

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
