## [Decision Letter]

Thank you for submitting your work entitled "Two-Subunit DNA Escort Mechanism and Inactive Subunit Bypass in an Ultra-Fast Ring ATPase" for peer review at *eLife*. Your submission has been favorably evaluated by Richard Losick (Senior editor), a Reviewing editor, and two reviewers.

The reviewers have discussed the reviews with one another and the Reviewing editor has drafted this decision to help you prepare a revised submission.

In this paper by Liu et al. the dsDNA translocation mechanism of the hexameric ring ATPase SpoIIIE is investigated. The authors use optical tweezers to monitor the translocation of the enzyme along DNA. In particular they study the bypass of short stretches of DNA with a neutral backbone as well as the influence of non-hydrolyzable ATP analogs. The study clearly shows that SpoIIIE contacts only one strand of the dsDNA during translocation and can traverse up to four consecutive MeP on one strand. However, five consecutive MeP hinder translocation of SpoIIIE. From this they conclude a two-subunit escort mechanism, where typically two adjacent subunits make contacts to the DNA. After stepping of one of the subunits, the DNA is moved by 2 bp providing engagement of an adjacent subunit in the translocation direction and disengagement of the trailing subunit. The experiments are well designed and carefully executed. The manuscript is clearly written and provides a consistent story. However, in contrast to previous studies where e.g. steps of the ring translocase from Phi29 were directly resolved, the present work relies on indirect evidence based on the length of backbone substitutions that can be overcome by the translocase. Though the argumentation of the authors is consistent and is further supported by the evaluation of the density of pauses originating from binding non-hydrolyzable ATP analogs, other and potentially more complex mechanisms cannot be convincingly ruled out. For example, the data does not eliminate another model where each SpoIIIE contacts one bp of dsDNA and five subunits are involved in DNA binding similar to the mechanism proposed for Rho and E1 helicases.

Essential revisions:

1) There is contradiction in what is said in the subsection “MeP Stepping Stone Constructs Suggest that SpoIIIE Has a Step Size of 2 bp” between the end of the last paragraph and the second paragraph. The authors say that the translocation/escort mechanism of Rho and E1 is similar to what they propose for SpoIIE. However, the two mechanisms are distinct. In Rho and E1, there are 5 consecutive subunits contacting the DNA and the step size is one bp whereas the proposed model of SpoIIIE has two consecutive subunits interacting with DNA. However, the data shown in this study does not eliminate that five consecutive subunits of SpoIIIE interact with the DNA and step size is one bp. The authors rule out this mechanism based on the stiffness of dsDNA (second paragraph). However, one can argue that the SpoIIIE ring can be distorted to enable interactions in a manner seen with Rho and E1.

2) Because the current data do not convincingly rule out other models, further support for the model is needed by challenging the model with additional stepping stone constructs. An interesting construct would be a 2 bp stepping stone, i.e. 4bp of neutral DNA followed by 2bp stepping stone and up to 4bp probe segment of neutral DNA. For a 2 bp stepping stone, the translocase should then be able to overcome a probe segment which would be 4 bp in length. A demonstration of this would significantly strengthen the current conclusions, in particular the 2 bp stepping which is in our opinion the most novel conclusion of the paper.

---

## [Author Response]

*[…] The experiments are well designed and carefully executed. The manuscript is clearly written and provides a consistent story. However, in contrast to previous studies where e.g. steps of the ring translocase from Phi29 were directly resolved, the present work relies on indirect evidence based on the length of backbone substitutions that can be overcome by the translocase. Though the argumentation of the authors is consistent and is further supported by the evaluation of the density of pauses originating from binding non-hydrolyzable ATP analogs, other and potentially more complex mechanisms cannot be convincingly ruled out. For example, the data does not eliminate another model where each SpoIIIE contacts one bp of dsDNA and five subunits are involved in DNA binding similar to the mechanism proposed for Rho and E1 helicases.*

Initially we did not consider a E1/Rho-like translocation mechanism as a potential model for SpoIIIE operation because this mechanism requires five consecutive motor subunits to contact five consecutive phosphate groups on the nucleic acid backbone. We reasoned that given the geometry of the SpoIIIE ring (300-degree arc for 5 subunits) and that of dsDNA (~170-degree arc for 5 consecutive backbone phosphates), the E1/Rho-like mechanism would require large distortions of the motor and/or dsDNA. However, even if we allowed for large motor/DNA distortions, the results of our study are inconsistent with the E1/Rho-like mechanism. A Discussion section and accompanying figures have now been added to the manuscript.

To summarize here, the crystal structures of the homo-hexameric helicases E1 and Rho strongly suggest a nucleic acid translocation mechanism where 5 ATPase subunits each contact consecutive phosphates on the ssDNA/ssRNA backbone and the turnover of one ATP molecule is coupled to translocation by one nucleotide. If SpoIIIE operated via the E1/Rho-like translocation mechanism (Figure 4Bv), it should have successfully traversed a double-stranded MeP insert of 4 bp, but not a dsMeP insert of 5 bp, as shown in the results presented in Figure 2.

How would this translocation mechanism cope with stepping stone constructs containing a 1-bp, 2-bp, 3-bp, or 4-bp MeP probe? Figure 5—figure supplement 1 illustrates two possible cases: (i) if the ATPase subunits have to fire in a strictly sequential order, a motor operating like E1/Rho would not cross a MeP stepping stone insert with a 1-bp probe, contrary to our observations (Figure 5); (ii) if the same ATPase subunit could fire out of order several times in a row, a motor employing a E1/Rho-like mechanism should traverse a MeP stepping stone insert with a 4-bp probe, inconsistent with our findings (Figure 5). We conclude that the results of our experiments rule out a E1/Rho-like translocation mechanism for SpoIIIE. In the revised manuscript we discuss the E1/Rho-like model and added Figure 5Bv and Figure 5—figure supplement 1 to illustrate its predictions.

*Essential revisions: 1) There is contradiction in what is said in the subsection “MeP Stepping Stone Constructs Suggest that SpoIIIE Has a Step Size of 2 bp” between the end of the last paragraph and the second paragraph. The authors say that the translocation/escort mechanism of Rho and E1 is similar to what they propose for SpoIIE. However, the two mechanisms are distinct. In Rho and E1, there are 5 consecutive subunits contacting the DNA and the step size is one bp whereas the proposed model of SpoIIIE has two consecutive subunits interacting with DNA. However, the data shown in this study does not eliminate that five consecutive subunits of SpoIIIE interact with the DNA and step size is one bp. The authors rule out this mechanism based on the stiffness of dsDNA (second paragraph). However, one can argue that the SpoIIIE ring can be distorted to enable interactions in a manner seen with Rho and E1.*

We initially compared the SpoIIIE model to the E1/Rho mechanism to highlight that in both cases multiple motor subunits can interact with the substrate. The other parallel we wanted to draw is that both SpoIIIE and E1/Rho models have “translocating” subunits and “escorting” subunits, but the actual number of escorting subunits is different (4 escorting subunits in E1/Rho and one escorting subunit in SpoIIIE). We thank the reviewers for pointing this out to us. The ambiguity in wording has been eliminated and now we also highlight the differences between our model and the E1/Rho mechanism.

*2) Because the current data do not convincingly rule out other models, further support for the model is needed by challenging the model with additional stepping stone constructs. An interesting construct would be a 2 bp stepping stone, i.e. 4bp of neutral DNA followed by 2bp stepping stone and up to 4bp probe segment of neutral DNA. For a 2 bp stepping stone, the translocase should then be able to overcome a probe segment which would be 4 bp in length. A demonstration of this would significantly strengthen the current conclusions, in particular the 2 bp stepping which is in our opinion the most novel conclusion of the paper.*

In general, constructs with stepping-stones of 2 bp (or longer) have a lower model-discriminating potential than constructs with 1-bp stepping-stones. For example, as shown in Figure 8, both the model with a 2-bp step size and the model with a 4-bp step size cannot traverse an insert with a 2-bp stepping-stone followed by a 3-bp MeP probe (but could have traversed the stepping-stone insert if the probe were one base-pair shorter).

Author response image 1.**DOI:**
http://dx.doi.org/10.7554/eLife.09224.015

In contrast, constructs with a 1-bp stepping-stone have the ability to discriminate between translocation models with 2-bp, 3-bp, 4-bp, and 5-bp step sizes as illustrated in Figure 5. We did initially consider inserts with a 2-bp stepping-stone, but opted for a design with 1-bp stepping-stone to avoid the sort of degenerate predictions illustrated in the diagram above. In addition, since SpoIIIE can slip under force and re-engage the DNA upstream of the slipping site (Figure 1, insert), constructs with a stepping-stone of 1 bp represent the most stringent crossing obstacles because 1 bp is the minimum foothold needed by the motor.